# A Coarse to Fine Detection Method for Prohibited Object in X-ray Images Based on Progressive Transformer Decoder*

## ABSTRACT

Currently, Transformer-based prohibited object detection methods in X-ray images appear constantly, but there are still some shortcomings such as poor performance and high computational complexity for prohibited object detection with heavily occlusion. Therefore, a coarse to fine detection method for prohibited object in X-ray images based on progressive Transformer decoder is proposed in this paper. Firstly, a coarse to fine framework is proposed, which includes two stages: coarse detection and fine detection. Through adaptive inference in stages, the computational efficiency of the model is effectively improved. Then, a position and class object queries method is proposed, which improves the convergence speed and detection accuracy of the model by fusing the position and class information of prohibited object with object queries. Finally, a progressive Transformer decoder is proposed, which distinguishes high and low score queries by increasing confidence thresholds, so that high-score queries are not affected by low-score queries in the decoding stage, and the model can focus more on decoding low-score queries, which usually correspond to prohibited object with severe occlusion. The experimental results on three public benchmark datasets (SIXray, OPIXray, HiXray) demonstrate that compared with the baseline DETR, the proposed method achieves the state-of-the-art detection accuracy with a 21.6% reduction in model computational complexity. Especially for prohibited objects with heavily occlusion, accurate detection can be carried out.

## CCS CONCEPTS

•Computing methodologies~Artificial intelligence~Computer vision~Computer vision problems~Object detection

## KEYWORDS

X-ray Image, Prohibited Object Detection, Coarse to Fine, Position and Class Object Queries, Progressive Transformer Decoder

## 1 INTRODUCTION

In airports, train stations, and port terminals, to effectively mitigate the security risks posed by controlled knives, explosives, firearms, chemicals, and other prohibited objects, X-ray machines are commonly used to inspect passengers' luggage, ensuring travel safety. Currently, the analysis of X-ray images primarily relies on manual inspection, which is susceptible to the influence of security personnel's condition and experience, and can easily lead to missed or false detections of prohibited objects, posing significant hidden dangers to passenger safety. To enhance the accuracy and efficiency of security checks, researchers have begun to explore various methods to achieve automatic and accurate detection of prohibited objects in X-ray images. However, compared with natural images, X-ray images exhibit many unique characteristics, such as the lack of inherent color, insufficient texture information, mutual occlusion between objects, clutter from surrounding items, and high similarity in imaging among some objects, which bring significant challenges and difficulties in detecting prohibited objects in X-ray images.

With the rapid development of deep learning, researchers have applied it to the detection of prohibited objects in X-ray images, significantly improving detection accuracy and efficiency. But in general, the detection performance for heavily occluded prohibited objects remains suboptimal. In recent years, the application of Transformer-based models in the detection of prohibited objects in X-ray images has achieved significant breakthroughs, showing great potential in addressing occluded prohibited object detection. Nevertheless, existing methods still need improvement in the following aspects:

(1) The detection speed is slow. The input tokens of the Transformer are sequential vectors, and in object detection tasks, the length of the token sequence is positively correlated with the size of the feature map. The length of the sequence has an exponential impact on the computational complexity, resulting in a significant increase in computational complexity. Therefore, it is necessary to research more efficient Transformer model structures or optimization methods to improve detection speed.

(2) Insufficient mining of object position and class information. Object detection methods based on Transformers rarely consider the object's position and class information when generating object queries, which not only makes network training difficult to converge quickly but also hinders further improvement of detection performance. Therefore, more effective methods for generating object queries need to be explored.

(3) The impact of low-score queries on detection results is ignored. In the Transformer decoder, low-score queries usually correspond to objects with severe occlusion in X-ray images, negatively affecting detection results. Therefore, it is necessary to design better learning strategies to reduce the impact of low-score queries to high-score queries during training and fully exploit the features of low-score queries to improve the detection accuracy of heavily occluded objects.

To solve the above problems, a coarse to fine prohibited object detection method in X-ray images based on Progressive

Transformer Decoder (PTD) is proposed in this paper, which effectively improves the detection accuracy of severe occlusion prohibited object by using the global modeling capability of Transformer. The main innovations include:

(1) A novel Coarse To Fine Framework (CTFF) is proposed for prohibited object detection, which performs staged adaptive inference. This framework not only enhances the computational efficiency of the model but also improves the detection accuracy.

(2) A novel method for generating object queries is introduced, namely the Position and Class Object Queries (PCOQ) method. This method integrates the position and class information of prohibited objects with object queries, enabling the model to locate the position and class of prohibited objects more accurately. This integration significantly enhances the model's convergence speed and detection accuracy.

(3) The PTD is proposed, which distinguishes high and low-score queries through increasing confidence threshold levels, thus shielding high-score queries from the influence of low-score queries during the decoding stage, and making the network more focused on decoding low-score queries, effectively improving the detection performance for severely occluded prohibited objects.

The remainder of the paper is arranged as follows: Section 2 introduces Related Work, and Section 3 introduces the method proposed in this paper. Section 4 is the Experimental Results and Analysis, and Section 5 is the Conclusion.

## 2  RELATED WORK

Prohibited object detection in X-ray images is a typical object detection problem. At present, object detection framework based on natural images is widely used. Existing prohibited object detection methods can be roughly divided into CNN-based methods and Transformer-based methods. The following are introduced separately.

## 2.1 CNN-based prohibited object detection in X-ray image

CNN-based prohibited object detection methods in X-ray images primarily include one-stage and two-stage detection approaches. One-stage methods generally employ architectures such as the YOLO [22] and SSD [15], which directly regress coordinates to determine the position of prohibited objects. Two-stage methods commonly use the R-CNN series [23] architecture, which detect prohibited objects based on region proposals.

Zhang et al. [39] adopts an improved Mask R-CNN [6], which integrates low-level features into high-level features to improve feature representation capability, and adopts SoftNMS [1] instead of traditional NMS [19] to overcome the missed detection problem of overlapping objects. Chen et al. [4] found that the atomic number Z information of objects with different materials is obviously different in X-ray images, which is very important to suppress the interference of irrelevant background information by mining material clues. Therefore, the authors proposed an atomic number Z Prior Guided Network (ZPGNet), which uses the atomic number Z information to mine objects and materials to reduce irrelevant background information. However, the authors

only pay attention to the atomic number Z information and does not fully consider the edge contour information of prohibited object, so it cannot solve the occlusion problem well.

To this end, Liu et al. [12] designed three mechanisms. Firstly, the authors proposed a scale interaction module, which makes the features of adjacent scales interact once or more times to enhance the perception ability of the model. Then, a cross-image weakly supervised semantic analysis model is designed, which uses collaborative attention mechanism to perceive similar and different objects, and breaks through the information bottleneck of isolated detection of a single image. Finally, they introduced a multi-task learning module, which optimizes the model at both global level and pixel level. In order to solve the occlusion problem and class imbalance problem at the same time, Liu et al. [14] proposed a dual multi-instance attention network named DMA-Net, which uses two different attention mechanisms, namely, image block-based and proposal-based multi-instance attention mechanisms, so as to extract the features of key instances. Based on image block attention mechanism, local information in X-ray image can be extracted and multiple instance features can be generated. The attention mechanism based on proposal can adaptively select important feature regions, so that the network can pay more attention to these regions.

X-ray images usually contain a large number of blank areas, so sending the whole image directly into the network for training is not only time-consuming, but also has poor detection effect. To solve this problem, Nguyen et al. [20] proposed a task-driven image cropping scheme called Task-Driven Cropping (TDC). It can adaptively clip the X-ray image, and quickly identify the importance of each pixel by using the depth feature extractor. Only the areas related to the detection task will be reserved and passed to the subsequent detectors.

CNN-based prohibited object detection methods in X-ray images are better at capturing the spatial location information of objects, providing advantages in object localization. However, these methods generate a large number of candidate boxes, for which Non-Maximum Suppression (NMS) is typically used to filter the candidate boxes. NMS itself is a greedy algorithm and cannot guarantee a globally optimal solution, potentially leading to missed or false detections. Moreover, the threshold for NMS needs to be manually set, usually based on experience and experimental results, making it difficult to generalize across different datasets and tasks.

## 2.2 Transformer-based prohibited object detection in X-ray image

In recent years, the Transformer has made significant progress in various fields such as object detection and semantic segmentation, and Transformer-based prohibited object detection methods in X-ray images [28] have also been emerging. DETR [2] is one of the earliest benchmark studies to apply the Transformer to object detection. This method first uses a CNN network as the backbone to extract image features, then flattens the output feature map to serve as a sequence input to the Transformer, and finally outputs

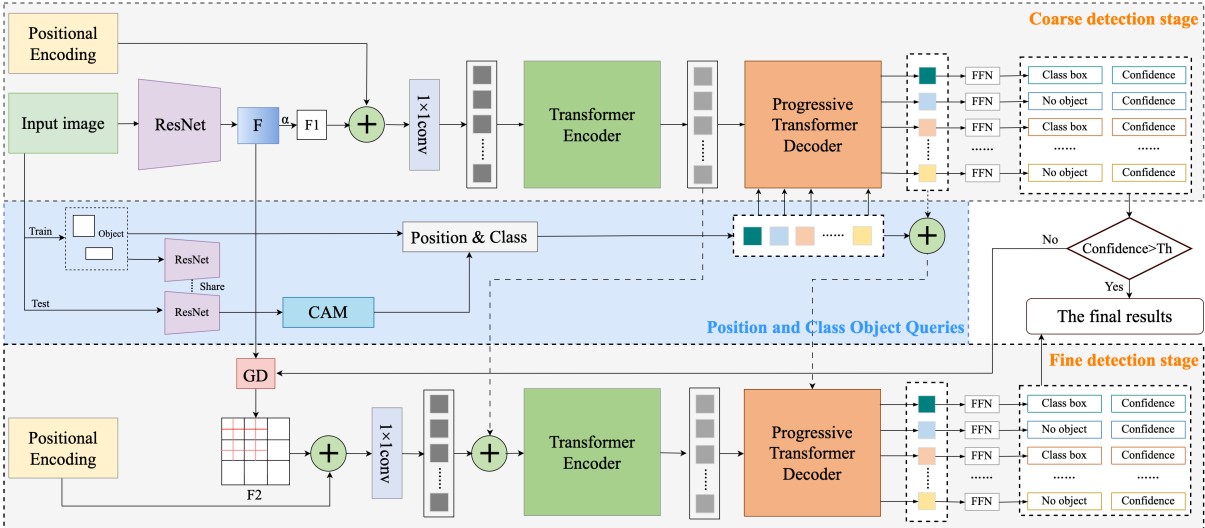

**Figure 1: The pipeline of the proposed method. It consists of the CTFF, PCOQ module, and PTD. In this structure, CTFF comprises two stages: coarse detection stage and fine detection stage. Given the characteristics of X-ray images, CTFF effectively reduces the computational complexity of the model. PCOQ provides relatively accurate initial values for object queries, which can accelerate network convergence. PTD is designed to handle prohibited object with severe occlusion. By segregating high and low score queries, the model can focus more on heavily occluded prohibited object, thereby enhancing the overall detection accuracy of the network. Note that F, F1, and F2 denote intermediate feature maps, CAM stands for Class Activation Map, and Th represents the confidence threshold.**

in parallel after processing by the Transformer, used for predictions to obtain the final results.

Besides DETR architecture, some researchers also tried to embed Transformer into object detection architecture based on CNN, and achieved good results. For example, Wang et al. [32] used an improved Transformer structure to optimize YOLO architecture. They replace the convolution layer in the original YOLO with a Transformer encoder, and add a Transformer decoder in the detection header to generate detection boxes and class predictions. This improved structure is helpful for the network to better deal with the relationship and context information between prohibited object, thus improving the detection accuracy. Wang et al. [31] proposed an object detection method called YOLOv4-T, which combines YOLOv4 with Transformer and optimizes the multi-scale object detection. The method has high robustness and good resistance to noise, illumination and other disturbances.

In order to solve the problem of class imbalance, Divya et al. [29] combined focus loss with Transformer, and implemented a weakly supervised location method based on Class Activation Map (CAM), which can help the model identify areas containing prohibited object. In order to make full use of X-ray images with multiple views, Brian et al. [7] proposed a prohibited object detection method based on multi-view Transformer, which can combine image information from multiple views to improve detection performance.

To improve the detection accuracy of small objects in X-ray images, Wang et al. [34] first added Transformer to the low-level of YOLOv5 [10] to make full use of shallow features. Then global attention mechanism is used to enhance the extracted features. Finally, an adaptive spatial feature fusion module is proposed to

enhance the utilization of feature map and strengthen the connection between pixels. Similarly, Wang et al. [33] proposed a method for detecting small prohibited objects in X-ray images called TB-YOLOv5. This method also embeds a Transformer module in the backbone of YOLOv5, uses an attention-enhanced BiFPN instead of the PANet structure in the neck, and integrates the BiFPN structure with coordinate attention to enhance the extraction capabilities of image features.

Compared with CNN-based methods for X-ray image detection, Transformer-based detection methods do not produce a large number of redundant boxes, thereby eliminating the need for NMS post-processing, which significantly reduces the possibility of missing objects. Additionally, since these methods do not require the pre-setting of anchors, they can largely avoid the impact of human factors on detection performance. As such, they are increasingly being applied to the detection of occluded prohibited objects, showing great potential. However, these methods also have high model complexity, which is not conducive to practical application deployment. To address this, CTFF is proposed for prohibited object detection in this paper, which not only reduces computational complexity but also enhances detection accuracy.

## 3 PROPOSED METHOD

To balance detection accuracy and speed, CTFF is proposed for prohibited object detection, as shown in Figure 1, which reduces the computational complexity by reducing the input token dimension of Transformer encoder. Then PCOQ is proposed to improve the object queries in order to speed up the convergence of the network and improve the detection accuracy. Finally, PTD

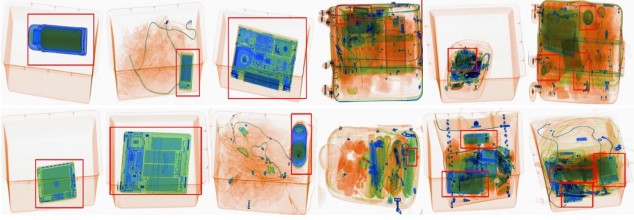

Simple prohibited object      Complex prohibited object

**Figure 2: Simple and Complex prohibited object in X-ray Images. It can be seen that for some complex prohibited object, it is difficult for even the human eye to distinguish.**

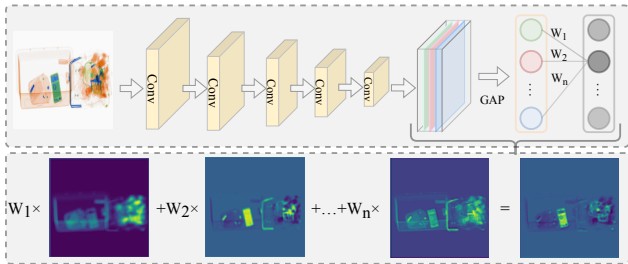

**Figure 3: Grid Decision Module. Based on the object positions during the coarse detection stage, different refinement rules are applied. For specific details, please refer to Section 3.1.**

**Figure 4: Calculation method of class activation map. By using a pre-trained prohibited object classification model, it can provide object queries with better prior information.**

is proposed to reduce the influence of low-score queries on high-score queries by progressively distinguishing queries. In the following sections, we will cover each part and the loss function separately.

## 3.1 Coarse to fine framework for prohibited object detection

Some simples and complex prohibited object examples are shown in Figure 2, which show that there is a significant variation in the backgrounds of X-ray images. Processing these images with a uniform approach would inevitably result in the wastage of resources. Inspired by this, a prohibited object detection framework that progresses from coarse to fine is proposed in this paper. Different methods are utilized for processing based on the complexity of the backgrounds in X-ray images.

The core of Transformer-based object detection framework is self-attention mechanism, in which each input element has a corresponding query, key and value vector, which is calculated by linear transformation. The attention scores are obtained by computing the dot product between the query and key vectors, and these scores are then used to perform a weighted average with the

**Table 1: Comparison of Computational Complexity of Different Components in Transformer.**

| Component | GFLOPs |
|-----------|--------|
| Encoder | 132G |
| Decoder | 22G |

value vectors, yielding the final output. This process can be viewed as a self-alignment of each element in the input sequence, enabling the model to better capture the relationships between elements at different positions within the sequence. As shown in Table 1, by analyzing the computational complexity of each component of Transformer, it can be seen that the maximum computational complexity of the whole method lies in the encoder part, and its computational complexity is usually positively correlated with the dimension of input token.

Based on the observations above, an adaptive prohibited object detection framework is proposed in this paper. For prohibited objects with simple imaging characteristics, lower-dimensional tokens are sufficient for effective representation. However, for objects with complex imaging characteristics, when low-dimensional tokens are inadequate to encapsulate all features, higher-dimensional tokens are utilized for representation.

The specific process is shown in Figure 1, which is mainly divided into two stages: coarse detection and fine detection. In the coarse detection stage, the input image is first fed into a ResNet50 model pre-trained on ImageNet for feature extraction. After obtaining the feature map $F$, the size of it is reduced by a reduction coefficient $\alpha$, where $\alpha=1/2$ in this paper. The reduced feature map is then merged with positional encoding, followed by processing through a 1×1 convolution. The processed features are used as tokens and sent to the Transformer for detection. The detection results are then evaluated, if the confidence level exceeds the threshold $Th$, it is concluded that the network has successfully detected the prohibited object, and this result is output as the final result. If the confidence level is below $Th$, it is determined that the network has failed to detect the prohibited object correctly, necessitating the second stage of fine detection.

After the coarse detection stage, the network has identified the approximate locations of the prohibited objects. Therefore, in the fine detection stage, the results from the coarse detection can be reused. The location information from these results is fed into the Grid Decision (GD) module, which further refines the feature map $F$ based on these results.

The specific workflow of the GD module is illustrated in Figure 3. Firstly, the feature map $F1$ is divided into 9 large regions arranged in a 3×3 grid. Subsequently, depending on their positions, these large regions are subdivided into multiple subregions, labeled A, B, C, and D. Finally, based on the locations identified during the coarse detection, different regions are selected for further refinement.

The subdivision rules are as follows:

- When the coarse detection object is located in Region A, refine Region D, the region containing the detection point, and the two adjacent regions of B+C.

- When the coarse detection object is located in Region B, refine Region D, the area containing the detection point, and the two adjacent regions of A.
- When the coarse detection object is located in Region C, refine Region D, the area containing the detection point, and the adjacent regions of A and B+C.
- When the coarse detection object is located in Region D, refine the adjacent regions of A and the regions of B+C. For instance, if the coarse detection object (indicated by a red dot in the figure) is located in D1, then refine A1, B1+C1+C2, B2+C3+C4, and D1+D2+D3+D4, resulting in the feature map F2.

After selecting the four refined regions, the remaining five regions are proportionally reduced according to the size of the refined subregions. These, along with the refined regions, are then processed by a 1×1 convolution to generate a sequence of tokens. Following the principle of reuse, these tokens are fused with the output from the Transformer encoder during the coarse detection stage, and the combined tokens are then reintroduced into the Transformer encoder for further processing. In the decoding stage, the outputs from the Transformer decoder during the coarse detection stage are also merged with the inputs from the fine detection stage to form the new object queries for input.

## 3.2 Position and Class Object Queries

In DETR architectures, object queries are often difficult to interpret and lack concrete physical meaning. During training, the lack of association between object queries and the actual features of objects leads to the network's inability to accurately narrow down the search area for objects, resulting in slower convergence and difficulty in achieving optimal detection accuracy.

To address this problem, the PCOQ is proposed in this paper, as shown in the middle part of Figure 1. This method utilizes CAM method to determine the approximate position and class of objects, providing a better positional prior and class information for subsequent object queries. In the training stage, the object regions from the input images are first extracted and trained using ResNet-18 to create a prohibited object classification network for X-ray images. Subsequently, the position and class vectors of the objects are used as object queries for training the decoder.

During the testing stage, input images are directly fed into the trained X-ray image prohibited object classification network to generate class activation maps. From these maps, the implied object position and class vectors are extracted, with subsequent steps consistent with those in the training stage. The CAM method used in this paper is illustrated in Figure 4.

As shown in the figure, the input image is sent to ResNet-18, and the global average pooling is performed on the feature map before the final output layer, which is taken as the weight of each layer. According to the different output categories, the weights of different classes are weighted and summed with feature map, and the class activation map of corresponding categories can be obtained. Through this simple network structure, the weight of the output layer can be projected onto the convolution feature map to determine the importance of the object region.

From the analysis above, it is evident that the PCOQ method proposed in this paper provides a better initial distribution for the network, allowing different queries to associate with specific positions and classes, which can facilitate faster convergence of the network.

## 3.3 Progressive Transformer Decoder

The Transformer architecture employs an encoder-decoder structure. Within the encoder, each point in the feature map interacts with surrounding points, allowing the network to learn a substantial amount of global semantic information. Subsequently, the output from the encoder is fed into the decoder to produce the final output results.

Research by Zheng et al. [42] demonstrates that when the output confidence of a query in the decoder exceeds a certain threshold, the majority of the Bounding Boxes (BBoxes) predicted are true positives. However, as the output confidence decreases, the number of false positives gradually increases. To effectively reduce the impact of low-score queries on high-score ones, this paper introduces a progressive Transformer decoder structure. This approach distinguishes queries by incrementally increasing confidence thresholds, isolating high-score queries for decoding in subsequent stages of the decoder. This method not only allows the decoder to focus more on extracting information from low-score queries but also prevents high-score queries from being influenced by these lower-score ones in later stages of decoding.

The structure of progressive Transformer decoder in this paper is shown in Figure 5 and comprises six decoders. In this structure, PE represents Positional Encoding, MSM represents the Multi-scale Self-attention Module, and MDM represents the Multi-scale Deformable-attention Module. QS represents the Query Selector, and QIE represents the Query Information Extractor.

The QS module is calculated as follows:

$$D_{t-1}^{high} = \left\{ b_i \mid s_i \geq s \cap b_i \in D_{t-1} \right\}, \tag{1}$$

$$D_{t-1}^{low} = D_{t-1} - D_{t-1}^{high}, \tag{2}$$

where $t$ denotes the stage sequence number, $D_{t-1}$ denotes the predicted set of all queries in $t$-1 stage, $D_{t-1}^{high}$ denotes the set of high-score queries, and $D_{t-1}^{low}$ denotes the set of low-score queries. $b_i$ denotes high score prediction box, $s_i$ denotes confidence, and $s$ denotes confidence threshold.

The QIE module is calculated as follows:

$$N(b_i) = \left\{ b_j \mid O(b_i, b_j) \geq \theta \right\}, b_i \in D_{t-1}^{low}, b_j \in D_{t-1}^{low}, \tag{3}$$

$$H(b_i) = f\left( \xi\left(b_i, N(b_i)\right)\right), b_i \in D_{t-1}^{high}, \tag{4}$$

$$R(b_i) = FC\left( MaxPool\left(H(b_i)\right) + F(q_i)\right), \tag{5}$$

where $N(b_i)$ denotes the neighbor of $b_i$, $b_j$ denotes the low-score prediction box, $O$ denotes the computational IoU operation, $\xi$ denotes the spatial position encoding, $f$ denotes the geometric relation of the spatial position encoding, $H$ denotes the generated geometric relation feature, $MaxPool$ denotes the maximum pooling operation, $F(q_i)$ denotes the feature generated by queries,

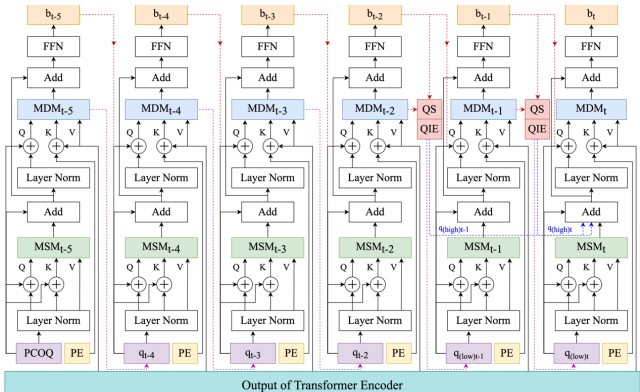

**Figure 5: Progressive Transformer Decoder. By segregating high and low score queries, the model can focus more on heavily occluded prohibited object.**

*FC* denotes the fully connected operation, and *R* denotes the extracted relation feature.

The outputs from the Transformer encoder, along with the object queries, are fed into the Progressive Transformer Decoder. After passing through three decoder stages, a large number of high-score queries have been obtained. Therefore, QS and QIE modules are added after the fourth and fifth stages. The specific workflow is as follows:

Firstly, the QS module is used to evaluate the output of the previous stage. If it exceeds a certain threshold, it is considered that this query is a high-score query and reserved. Otherwise, it is regarded as a low-score query. Then, the low-score query is sent into the QIE module, and it is evaluated whether the high-score query is contained around the low-score query. If so, the low-score query is discarded to avoid redundant detections. Otherwise, the low-score query will continue to be fed into the subsequent stage. Finally, the outputs of high-score queries from the third and fourth decoders are combined with the output from the fifth decoder and sent to the sixth decoder as final prediction result.

After the progressive decoding process, the model primarily focuses on decoding low-score queries, which typically correspond to the most severely occluded objects. Therefore, combined with the Transformer's global attention mechanism, this method effectively addresses the detection of prohibited object with significant occlusion.

## 3.4 Loss

The loss calculation in this paper consists of two parts: one is object detection box matching loss, and the other is the loss between matched BBoxes. It can be expressed as:

$$L_{total}\left(y,\hat{y}\right) = \min \sum_{t-1}^{N} L_{hungarian}\left(y_i, \hat{y}_{\sigma(i)}\right) | L_{match}\left(y_i, \hat{y}_{\sigma(i)}\right), \quad (6)$$

The detection box matching loss is expressed as:

$$L_{match}\left(y_i, \hat{y}_{\sigma(i)}\right) = -\text{I}_{\{c_i \neq \varnothing\}}\hat{p}_{\sigma(i)}\left(c_i\right) + \text{I}_{\{c_i \neq \varnothing\}}L_{box}\left(b_i, \hat{b}_{\sigma(i)}\right), \quad (7)$$

$$L_{match}\left(b_i, \hat{b}_{\sigma(i)}\right) = \lambda_{IoU}L_{IoU}\left(b_i, \hat{b}_{\sigma(i)}\right) + \lambda_{L1}\left\|b_i - \hat{b}_{\sigma(i)}\right\|_1, \quad (8)$$

where $c_i$ denotes the class of the *i*-th BBox, and $\text{I}_{\{c_i \neq \varnothing\}}$ denotes a value of 1 when the class is non-background. $\sigma(i)$ denotes the index of the *i*-th prediction box, and $\hat{p}$ denotes the class probability of prediction. $b_i$ denotes the coordinates of Ground Truth, and $\hat{b}$ denotes the coordinates of predicted BBox. $\lambda_{IOU}$ and $\lambda_{L1}$ are two hyperparameters, and $L_{IOU}$ represents GIoU loss [24].

The loss between matched BBox can be expressed as:

$$L_{Hungarian}\left(y_i, \hat{y}_{\sigma(i)}\right) = \sum_{i=1}^{N}\left[-\log \hat{p}_{\sigma(i)}\left(c_i\right) + \text{I}_{\{c_i \neq \varnothing\}}L_{box}\left(b_i, \hat{b}_{\sigma(i)}\right)\right], \quad (9)$$

where *N* denotes the number of BBox matched.

## 4 EXPERIMENTAL RESULTS AND ANALYSIS

### 4.1 Datasets and Evaluation Criteria

In order to verify the performance of the proposed method, experiments are carried out on three common benchmark datasets, namely SIXray, OPIXray and HiXray. And mean Average Precision (mAP) are often utilized as the evaluation metrics in object detection tasks[9, 40], thus, we also strictly follow these metrics in our experiments.

Next, the experimental results and analysis are introduced.

### 4.2 Implementation Details

The hardware configuration of the experiment is: 11th Gen Intel (R) Core (TM) i7-11700K @ 3.60 GHz, the graphics card is Quadro RTX 6000, 24G video memory and 32G memory. The network is built, trained and tested on PyTorch platform.

The ResNet-50 pre-trained on ImageNet as the backbone network in this paper. The model trains 50 epoch, and the learning rate becomes one tenth of the original when epoch is 40. Adam optimizer is used in the training process, and the initial learning rate is set to 2e-4, the weight decay is set to e-4, and the batch size is 1. No data enhancement operation is carried out.

### 4.3 Comparison with State-of-the-art Methods

In order to verify the effectiveness of the proposed method, we compare it with many existing object detection methods, including CNN-based methods and Transformer-based methods, among which CNN-based methods include: One-stage methods: YOLO v5 [10]*, FCOS[27]*, FA [35], MCIA-FPN [30], POD[17], Chang et al. [3], FA [35], ATSS+LAreg [41], Assails [41], ZPGNet [4], DOAM [36], LIM[26] and YOLO X[5]. Two-stage methods: Faster R-CNN [23]*, Grid R-CNN [16]*, Dynamic R-CNN [38], Dh_Faster R-CNN [37]*, Sparse R-CNN [25]* and DetectoRS [21]*. Transformer-based methods: Deformable DETR[44]*, Conditional DETR[18]*, DN DETR[11]*, DAB DETR[13]*, H Deformable DETR[8]* and Focus-DETR [43].

Table 2 shows the experimental results using different object detection methods on three datasets, where * represents the result reproduced under the same experimental conditions. The optimal results are represented in red font, and the suboptimal results are represented in blue font. It can be seen from the table that the proposed method achieves the best detection performance across

**Table 2: Performance comparison results using different object detection methods on SIXray, OPIXray, and HiXray datasets.**

| Type | | Method | Year | Backbone | Datasets | | | Inference Time(s) | GFLOPs | Parameters(MB) |
|---|---|---|---|---|---|---|---|---|---|---|
| | | | | | SIXray | OPIXray | HiXray | | | |
| CNN | Two-stage | Faster R-CNN [23]* | 2016 | ResNet-50 | 82.8 | 81.9 | / | 0.0343 | 210.47 | 41.09 |
| | | Grid R-CNN [16]* | 2019 | ResNet-50 | 83.7 | 83.6 | / | 0.0861 | 328.84 | 64.32 |
| | | Dynamic R-CNN [38]* | 2020 | ResNet-50 | 83.3 | 81.1 | / | 0.0439 | 215.44 | 41.14 |
| | | Dh_Faster R-CNN [37]* | 2020 | ResNet-50 | 83.7 | 82.6 | / | 0.2568 | 490.02 | 47.12 |
| | | Sparse R-CNN [25]* | 2021 | ResNet-50 | 81.3 | 81.8 | / | 0.1010 | 236.31 | 125.06 |
| | | DetectoRS [21]* | 2021 | ResNet-50 | 83.1 | 82.0 | / | 0.0588 | 228.01 | 93.47 |
| | | POD-F-R [17] | 2023 | ResNet-50 | 86.1 | 84.9 | / | 0.1498 | 334.12 | 118.23 |
| | | POD-F-X [17] | 2023 | ResNeXt-50 | 86.9 | 86.1 | / | 0.1563 | 337.44 | 119.67 |
| | One-stage | YOLO v5 [10]* | 2021 | CSPDarknet-53 | 86.7 | 87.9 | 81.7 | 0.0415 | 101.32 | 44.10 |
| | | FA [35] | 2021 | ResNet-50 | 85.8 | 87.7 | / | / | / | / |
| | | MCIA-FPN [30] | 2022 | ResNet-101 | 85.23 | 85.89 | / | / | / | / |
| | | YOLOX-L[5] | 2021 | Modified CSP v5 | / | / | 84.1 | / | / | / |
| | | ZPGNet [4] | 2023 | DarkNet-53 | / | 85.4 | 84.4 | / | / | / |
| | | POD-Y [17] | 2023 | CSPDarknet-53 | 90.4 | 90.9 | / | 0.0422 | 108.10 | 47.19 |
| Transformer | | Deformable DETR [44]* | 2020 | ResNet-50 | 89.94 | 89.76 | 80.77 | 0.0799 | 163.21 | 41.16 |
| | | Conditional DETR [18]* | 2021 | ResNet-50 | 90.14 | 89.68 | 83.18 | 0.0448 | 110.22 | 44.43 |
| | | DN DETR [11]* | 2022 | ResNet-50 | 89.88 | 90.07 | 80.32 | 0.0646 | 214.91 | 47.41 |
| | | DAB DETR [13]* | 2022 | ResNet-50 | 90.05 | 90.29 | 81.33 | 0.0604 | 207.12 | 45.67 |
| | | H D DETR [8]* | 2023 | ResNet-50 | 90.31 | 89.10 | 80.26 | 0.0796 | 218.90 | 47.98 |
| | | Focus-DETR [43]* | 2023 | ResNet-50 | 90.21 | 90.01 | 82.45 | 0.0601 | 179.48 | 47.84 |
| | | Ours | / | ResNet-50 | 92.39 | 92.47 | 85.75 | 0.0545 | 173.85 | 41.28 |

all datasets. On the three public benchmark datasets, the mAP values are higher than those of existing methods by 1.99% to 11.09%, 1.57% to 11.37%, and 1.35% to 14.35%, reaching 92.39%, 92.47%, and 85.75%, respectively. The comparison results indicate that Transformer-based methods generally achieve superior detection performance compared with CNN-based methods, especially for objects with severe occlusion. However, due to the lack of inductive bias, Transformer-based methods usually rely on large-scale annotation data. In HiXray dataset, compared with other categories, the number of NL category is relatively small, so most Transformer-based methods have poor detection effect. However, PCOQ can give object queries a more accurate initial value in advance, so it can still carry out better detection even if the number of labeled data is small.

To validate the efficiency of the proposed method, we compared its computational complexity and parameters with other detection methods. It is evident that this method has significant advantages in terms of computational complexity and model parameters compared with others. In terms of inference speed, the method surpasses most DETR-like methods without requiring additional model parameters. Considering the accuracy of the model, it is evident that the proposed method achieves an effective trade-off between precision and speed.

## 4.4 Ablation Studies

To further validate the effectiveness of each component within the proposed method, we conducted ablation studies. Table 3 shows the comparative results obtained by incorporating different modules across multiple datasets. Where baseline refers to the original DETR model.

It can be clearly seen that the computational complexity of the model was effectively reduced after using the CTFF. The

inference time decreased from 0.0834 seconds to 0.0510 seconds, a substantial reduction of 38.85%. Furthermore, FLOPs were reduced from 221.81G to 167.45G, achieving a computational saving of 24.51%, and the model's parameter size decreased from 46.70MB to 41.28MB, saving 14.69%. Despite these reductions, thanks to our feature reuse mechanism, the overall detection performance was minimally impacted, with an average decrease of just 0.29%.

After incorporating PCOQ and PTD, there was a significant improvement in detection accuracy, particularly with PTD, which increased accuracy by 1.26% on the SIXray dataset, 1.18% on the OPIXray dataset, and 1.62% on the HiXray dataset. Following the implementation of PCOQ, accuracy also increased by 0.95%, 1.03%, and 1.24%, respectively. Ultimately, with a 21.6% reduction in computational complexity, the method achieved optimal detection accuracy across all three datasets, thoroughly demonstrating its effectiveness. The OPIXray and HiXray datasets, which are more complex, were most affected by the CTFF, showing the largest declines in detection accuracy. However, thanks to the positional and class information provided by PCOQ, the models were still able to achieve the best detection accuracy.

To further investigate the various modules proposed in this paper, we conducted more detailed ablation studies using the SIXray dataset as an example.

*4.4.1 Effect of different CTFF parameters on model performance.* To explore the impact of different stages on detection performance in CTFF, the performance before and after the incorporation of the fine detection stage are compared, as shown in Table 4. In this table, "Baseline" refers to the results after removing the CTFF, "Coarse" indicates the coarse detection stage, and "Fine" denotes the fine detection stage. The data reveals that when only the coarse detection is performed, there is a

**Table 3: The effect of different components of the proposed method on detection performance.**

| Baseline | CTFF | PCOQ | PTD | SIXray | OPIXray | HiXray | Inference Time(s) | GFLOPs | Parameters(MB) |
|---|---|---|---|---|---|---|---|---|---|
| √ | | | | 90.32 | 90.61 | 83.28 | 0.0904 | 221.81 | 46.70 |
| √ | √ | | | 90.18 | 90.26 | 82.89 | 0.0510 | 167.45 | 37.84 |
| √ | √ | √ | | 91.13 | 91.29 | 84.13 | 0.0522 | 170.11 | 41.01 |
| √ | √ | √ | √ | 92.39 | 92.47 | 85.75 | 0.0545 | 173.85 | 41.28 |

**Table 4: The impact of different stages on detection performance in CTFF.**

| Baseline | Coarse | Fine | mAP | Inference Time(s) | GFLOPs |
|---|---|---|---|---|---|
| √ | | | 92.50 | 0.0958 | 229.21 |
| √ | √ | | 90.91 | 0.0424 | 151.23 |
| √ | √ | √ | 92.39 | 0.0545 | 173.85 |

**Table 5: The impact of different backbone on model performance in PCOQ.**

| Backbone | Classification accuracy | mAP | Inference Time(s) | GFLOPs |
|---|---|---|---|---|
| ResNet-18 | 94.0 | 92.39 | 0.0545 | 173.85 |
| ResNet-34 | 94.2 | 92.40 | 0.0581 | 175.61 |
| ResNet-50 | 94.1 | 92.40 | 0.0600 | 176.20 |
| ResNet-101 | 94.2 | 92.41 | 0.0639 | 180.78 |

**Table 6: The impact of the number of QS-QIE modules on model performance.**

| Number of QS-QIE modules | mAP | Inference Time(s) | GFLOPs |
|---|---|---|---|
| 0 | 91.13 | 0.0522 | 170.11 |
| 1 | 91.85 | 0.0531 | 171.91 |
| 2 | 92.39 | 0.0545 | 173.85 |
| 3 | 92.36 | 0.0573 | 175.79 |

significant drop in model accuracy, but the inference time is greatly reduced, with a decrease of up to 56%. After incorporating the fine detection stage, which adds more feature information and utilizes feature reuse, the model is able to capture a richer set of useful features, resulting in a substantial improvement in accuracy, with the mAP increasing by 1.48%.

*4.4.2 Effect of different backbone in PCOQ on model performance.* In the PCOQ module, we initially trained a prohibited object classification network using X-ray images to evaluate the impact of different backbones on detection accuracy. We conducted comparative experiments with ResNet-18, ResNet-34, ResNet-50, and ResNet-101, as shown in Table 5. The data indicates that the differences in classification accuracy among these backbones are very minimal, ranging only between 0.1% and 0.2%. Such slight variations have a negligible impact on the final detection outcomes. For the datasets we used, although switching to a larger capacity classification model could yield slightly better performance, it would come at the cost of increased computational complexity and model parameters. Therefore, considering the trade-off between speed and accuracy, ResNet-18 was chosen as the backbone for PCOQ in this paper.

*4.4.3 Effect of different PTD parameters on model performance.* After passing through the Transformer encoder, the input image is encoded into more abstract feature vectors, which

need to be decoded to generate the final detection results. To determine the appropriate number of QS and QIE modules, we conducted ablation experiments. For ease of understanding, we refer to the serial combination of these two modules as the QS-QIE module. When the number of QS-QIE modules is zero, it indicates that there is no segregation of high and low score queries. Since the segregation of queries occurs in the final stages of decoding, having one QS-QIE module means incorporating it before the sixth layer of the Transformer decoder. Two modules indicate insertion before the fifth and sixth layers of the Transformer decoder, and three modules indicate placement before the fourth, fifth, and sixth layers. The experimental results are shown in Table 6.

The data reveals that model performance is the worst when not using any QS-QIE modules. This is because the Transformer decoder typically contains a large number of low-score queries mixed with high-score queries, introducing significant noise into the model's decoding process for high-score queries. As the number of QS-QIE modules increases, high and low score queries begin to be segregated, and the model focuses more on decoding low-score queries, thereby significantly improving detection accuracy. When the number reaches two, the mAP increases by 1.26%. However, more QS-QIE modules are not always better. When the modules are added before the fourth layer of the Transformer decoder, the features from the Transformer encoder are not fully decoded yet, leading to inadequate judgment capability for distinguishing high and low score queries, thus slightly decreasing the mAP.

## 5 CONCLUSIONS

A coarse to fine detection method for prohibited object in X-ray images based on progressive Transformer decoder is proposed in this paper. Through comprehensive analysis of a vast array of experimental results, the following conclusions can be drawn: (1) The background complexity of X-ray images is very different, so different computing resources are given according to the difficulty of samples, which is conducive to both detection accuracy and speed. (2) Introducing as much prior knowledge about prohibited items as possible during the initialization of object queries can help the model converge more rapidly and enhance detection accuracy. (3) The Transformer decoding stage contains a large number of low-score queries, which usually correspond to the most occluded objects. By using a mechanism to separate high and low-score queries, the training process can minimize the impact of low-score queries on high-score ones, allowing the model to dedicate more resources to decoding low-score queries and thus enhancing the detection accuracy for heavily occluded prohibited objects.

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
