# OpenReview forum: "A Coarse to Fine Detection Method for Prohibited Object in X-ray Images Based on Progressive Transformer Decoder"
_acmmm.org/ACMMM/2024/Conference — MM2024 Poster_

### Official Review · Reviewer_rhxi · 2024-05-23

**Rating:** 4
**Confidence:** 3

**Summary:**

The paper presents a novel method for detecting prohibited objects in X-ray images using a progressive Transformer decoder. This method addresses the challenges of high computational complexity and poor detection performance, particularly for heavily occluded objects. It introduces a two-stage framework, coarse detection followed by fine detection, to enhance computational efficiency and detection accuracy. The method also incorporates Position and Class Object Queries (PCOQ) to integrate positional and class information, accelerating model convergence. Furthermore, the Progressive Transformer Decoder (PTD) isolates high-score queries from low-score ones, allowing the model to focus on accurately decoding occluded objects. Experimental results on public benchmark datasets demonstrate significant improvements in detection accuracy and computational efficiency over existing methods.

**Strengths:**

which integrates positional and class information with object queries, represents a significant advancement over traditional detection methods.

2.The paper demonstrates a high level of technical correctness. The authors meticulously explain the design and implementation of their proposed method, including detailed descriptions of the Progressive Transformer Decoder (PTD) and the PCOQ method. The mathematical formulations provided, such as the calculation of attention scores and the design of the progressive decoder, are robust and well-founded. For instance, the PTD's approach to distinguishing high and low-score queries using increasing confidence thresholds is both theoretically sound and practically effective.

3. The paper includes an extensive evaluation of the proposed method using three public benchmark datasets (SIXray, OPIXray, HiXray). The results demonstrate that the method achieves state-of-the-art detection accuracy while reducing model computational complexity by 21.6% compared to the baseline DETR. The evaluation metrics used, such as mean Average Precision (mAP), provide a clear and objective measure of performance. Additionally, the paper conducts ablation studies to validate the effectiveness of each component within the proposed method, further reinforcing the credibility of the results.

**Limitations:**

1. Considering the research area of this paper, the performance of devices is often limited in practical scenarios. This paper could provide inference speeds on different devices to further illustrate its practicality.
2. Table 2 shows the comparison results of different methods. It is clear that single-stage methods perform better than two-stage methods. Could the authors further explain why the two-stage method in this paper performs better?
3. While the Progressive Transformer Decoder (PTD) effectively handles low-score queries by distinguishing them from high-score queries, the process of progressively increasing confidence thresholds may introduce biases. For instance, there might be cases where true positives are misclassified as low-score queries due to their inherent characteristics, such as size or occlusion level. This could potentially lead to missed detections. Further refinement of the query handling mechanism is needed to balance the trade-off between eliminating false positives and retaining true positives.

**Suitability:**

3

---

### Official Review · Reviewer_KSy3 · 2024-05-28

**Rating:** 3
**Confidence:** 3

**Summary:**

The paper proposes a new method for detecting prohibited objects in X-ray images using a progressive Transformer decoder. This method includes a coarse to fine detection framework, a position and class object queries method, and a progressive decoder that prioritizes decoding low-score queries for heavily occluded objects. Experimental results on three benchmark datasets show that the proposed method improves detection accuracy while reducing computational complexity, especially for heavily occluded prohibited objects.

**Strengths:**

- This paper is well written with clear representation and concise illustration. For example, Figure 1 is very informative and concise, and I could clearly get the overall idea at first glance.
- Every module is well ablated. Overall performance on three datasets is good.

**Limitations:**

-The submitted PDF is not the specified text template.
-From Table 3, it can be seen that the CTFF module contributes negatively to the overall solution.
-The coarse-to-fine strategy in detection is not insightful and dose not provide takeaways for the community.

**Suitability:**

2

---

### Official Review · Reviewer_WWb2 · 2024-06-03

**Rating:** 4
**Confidence:** 2

**Summary:**

This paper aims to detect prohibited objects in x-ray images using a coarse-to-fine detection (CTTF) framework, along with two modules: Position and Class Object Queries (PCOQ) and Progressive Transformer Decoder (PTD). The method specifically aims to improve the accuracy of detecting heavily occluded prohibited objects while maintaining a low computation cost. It compares performance on multiple datasets with several previous methods and conducts ablation studies to evaluate the effectiveness of each component.

**Strengths:**

**Performance**: The accuracy of the proposed method is better than other methods listed, and it surpasses the baseline method with lower inference time and lower computation cost.

**Ablation Study**: It demonstrates the effectiveness of each proposed module with a detailed ablation study.

**Technical Correctness**: The idea of a coarse-to-fine framework and feature aggregation (the inclusion of class prior and object location prior, the integration of features from the coarse detection stage into the fine detection stage) is easy to understand and intuitively makes sense if properly used.

**Limitations:**

**Clarity**: The figures could be improved. For example, Figure 1 lacks a complete legend, e.g. there are two dashed arrows (running from the coarse detection stage to the fine detection stage) that are not labeled. I would assume the dashes indicate selected embeddings instead of all input embeddings, but this is not clear. Another example is that Figure 4 contradicts the text. GAP indicates it is pooling features from the whole image, while the text describes CAM as gathering features for each individual object proposal.

**Insufficient Explanation**: The speedup compared to the baseline DETR needs clarification. In Tables 3 and 4, the proposed method is significantly faster than the baseline, but the reason is not highlighted in the text. Is it because the early acceptance of high-scored objects reduces the number of input embeddings to the following decoders? Or does large patch division reduce the number of input embeddings to the whole network in the coarse detection stage?

**Suitability:**

2

---

### Meta-Review · Area_Chair_ugM8 · 2024-07-02

**Recommendation:** Accept (Poster)
**Confidence:** 5

**Metareview:**

After rebuttal, this paper received acceptance recommendations from all three reviewers. AC is happy to recommend acceptance and invites the authors to incorporate the materials from the rebuttal to the camera ready. Moreover, while reviewers appreciated the results and extensive experiments of this paper, one of them still leaves two concerns about the inference time and the potential bias problem. The authors are also supposed to include a discussion of these concerns in the final version.